# META-LEARNING CURIOSITY ALGORITHMS

**Ferran Alet**[∗]**, Martin F. Schneider**[∗]**, Tomás Lozano-Pérez & Leslie Pack Kaelbling**
Computer Science and Artificial Intelligence Laboratory
Massachusetts Institute of Technology
Cambridge, MA 02139, USA
`{alet,martinfs,tlp,lpk}@mit.edu`

## ABSTRACT

We hypothesize that curiosity is a mechanism found by evolution that encourages meaningful exploration early in an agent's life in order to expose it to experiences that enable it to obtain high rewards over the course of its lifetime. We formulate the problem of generating curious behavior as one of meta-learning: an outer loop will search over a space of curiosity mechanisms that dynamically adapt the agent's reward signal, and an inner loop will perform standard reinforcement learning using the adapted reward signal. However, current meta-RL methods based on transferring neural network weights have only generalized between very similar tasks. To broaden the generalization, we instead propose to meta-learn algorithms: pieces of code similar to those designed by humans in ML papers. Our rich language of programs combines neural networks with other building blocks such as buffers, nearest-neighbor modules and custom loss functions. We demonstrate the effectiveness of the approach empirically, finding two novel curiosity algorithms that perform on par or better than human-designed published curiosity algorithms in domains as disparate as grid navigation with image inputs, acrobot, lunar lander, ant and hopper.

## 1 INTRODUCTION

When a reinforcement-learning agent is learning to behave, it is critical that it both explores its domain and exploits its rewards effectively. One way to think of this problem is in terms of *curiosity* or *intrisic motivation*: constructing reward signals that augment or even replace the extrinsic reward from the domain, which induce the RL agent to explore their domain in a way that results in effective longer-term learning and behavior (Pathak et al., 2017; Burda et al., 2018; Oudeyer, 2018). The primary difficulty with this approach is that researchers are hand-designing these strategies: it is difficult for humans to systematically consider the space of strategies or to tailor strategies for the distribution of environments an agent might be expected to face.

We take inspiration from the curious behavior observed in young humans and other animals

Figure 1: Our RL agent is augmented with a *curiosity module*, obtained by meta-learning over a complex space of programs, which computes a pseudo-reward $\widehat{r}$ at every time step.

and hypothesize that curiosity is a mechanism found by evolution that encourages meaningful exploration early in an agent's life. This exploration exposes it to experiences that enable it to learn to obtain high rewards over the course of its lifetime. We propose to formulate the problem of generating curious behavior as one of meta-learning: an outer loop, operating at "evolutionary" scale will search over a space of algorithms for generating curious behavior by dynamically adapting the

---

[∗]Equal contribution.

agent's reward signal, and an inner loop will perform standard reinforcement learning using the adapted reward signal. This process is illustrated in figure 1; note that the aggregate agent, outlined in gray, has the standard interface of an RL agent. The inner RL algorithm is continually adapting to its input stream of states and rewards, attempting to learn a policy that optimizes the discounted sum of proxy rewards $\sum_{k\geq 0} \gamma^k \widehat{r}_{t+k}$. The outer "evolutionary" search is attempting to find a program for the curiosity module, so as to optimize the agent's lifetime return $\sum_{t=0}^{T} r_t$, or another global objective like the mean performance on the last few trials.

In this meta-learning setting, our objective is to find a curiosity module that works well given a distribution of environments from which we can sample at meta-learning time. Meta-RL has been widely explored recently, in some cases with a focus on reducing the amount of experience needed by initializing the RL algorithm well (Finn et al., 2017; Clavera et al., 2019) and, in others, for efficient exploration (Duan et al., 2016; Wang et al., 2017). The environment distributions in these cases have still been relatively low-diversity, mostly limited to variations of the same task, such as exploring different mazes or navigating terrains of different slopes. We would like to discover curiosity mechanisms that can generalize across a much broader distribution of environments, even those with different state and action spaces: from image-based games, to joint-based robotic control tasks. To do that, we perform meta-learning in a rich, combinatorial, open-ended space of programs.

This paper makes three novel contributions.

**We focus on a regime of meta-reinforcement-learning in which the possible environments the agent might face are dramatically disparate and in which the agent's lifetime is very long.** This is a substantially different setting than has been addressed in previous work on meta-RL and it requires substantially different techniques for representation and search.

**We propose to do meta-learning in a rich, combinatorial space of programs rather than transferring neural network weights.** The programs are represented in a *domain-specific language* (DSL) which includes sophisticated building blocks including neural networks complete with gradient-descent mechanisms, learned objective functions, ensembles, buffers, and other regressors. This language is rich enough to represent many previously reported hand-designed exploration algorithms. We believe that by performing meta-RL in such a rich space of mechanisms, we will be able to discover highly general, fundamental curiosity-based exploration methods. This generality means that a relatively computationally expensive meta-learning process can be amortized over the lifetimes of many agents in a wide variety of environments.

**We make the search over programs feasible with relatively modest amounts of computation.** It is a daunting search problem to find a good solution in a combinatorial space of programs, where evaluating a single potential solution requires running an RL algorithm for up to millions of time steps. We address this problem in multiple ways. By including environments of substantially different difficulty and character, we can evaluate candidate programs first on relatively simple and short-horizon domains: if they don't perform well in those domains, they are pruned early, which saves a significant amount of computation time. In addition, we predict the performance of an algorithm from its structure and operations, thus trying the most promising algorithms early in our search. Finally, we also monitor the learning curve of agents and stop unpromising programs before they reach all $T$ environment steps.

We demonstrate the effectiveness of the approach empirically, finding curiosity strategies that perform on par or better than those in published literature. Interestingly, the top 2 algorithms, to the best of our knowledge, had not been proposed before, despite making sense in hindsight. We conjecture the first one (shown in figure 3) is deceptively simple and that the complexity of the other one (figure 10 in the appendix) makes it relatively implausible for humans to discover.

## 2 PROBLEM FORMULATION

### 2.1 META-LEARNING PROBLEM

Let us assume we have an agent equipped with an RL algorithm (such as DQN or PPO, with all hyperparameters specified), $\mathcal{A}$, which receives states and rewards from and outputs actions to an environment $\mathcal{E}$, generating a stream of experienced transitions $e(\mathcal{A}; \mathcal{E})_t = (s_t, a_t, r_t, s_{t+1})$. The agent continually learns a policy $\pi(t) : s_t \rightarrow a_t$, which will change in time as described by algorithm $\mathcal{A}$;

so $\pi(t) = \mathcal{A}(e_{1:t-1})$ and thus $a_t \sim \mathcal{A}(e_{1:t-1})(s_t)$. Although this need not be the case, we can think of $\mathcal{A}$ as an algorithm that tries to maximize the discounted reward $\sum_i \gamma^i r_{t+i}, \gamma < 1$ and that, at any time-step $t$, always takes the greedy action that maximizes its estimated expected discounted reward.

To add exploration to this policy, we include a *curiosity module* $\mathcal{C}$ that has access to the stream of state transitions $e_t$ experienced by the agent and that, at every time-step $t$, outputs a proxy reward $\widehat{r}_t$. We connect this module so that the original RL agent receives these modified rewards, thus observing $e(\mathcal{A}, \mathcal{C}; \mathcal{E})_t = (s_t, a_t, \widehat{r}_t = \mathcal{C}(e_{1:t-1}), s_{t+1})$, without having access to the original $r_t$. Now, even though the inner RL algorithm acts in a purely exploitative manner with respect to $\widehat{r}$, it may efficiently explore in the outer environment.

Our overall goal is to design a curiosity module $\mathcal{C}$ that induces the agent to maximize $\sum_{t=0}^{T} r_t$, for some number of total time-steps $T$ or some other global goal, like final episode performance. In an episodic problem, $T$ will span many episodes. More formally, given a single environment $\mathcal{E}$, RL algorithm $\mathcal{A}$, and curiosity module $\mathcal{C}$, we can see the triplet (environment, curiosity module, agent) as a dynamical system that induces state transitions for the environment, and learning updates for the curiosity module and the agent. Our objective is to find $\mathcal{C}$ that maximizes the expected original reward obtained by the composite system in the environment. Note that the expectation is over two different distributions at different time scales: there is an "outer" expectation over environments $\mathcal{E}$, and in "inner" expectation over the rewards received by the composite system in that environment, so our final objective is:

$$\max_{\mathcal{C}} \left[ \mathbb{E}_{\mathcal{E}} \left[ \mathbb{E}_{r_t \sim e(\mathcal{A}, \mathcal{C}; \mathcal{E})} \left[ \sum_{t=0}^{T} r_t \right] \right] \right] \quad .$$

## 2.2 PROGRAMS FOR CURIOSITY

In science and computing, mathematical language has been very successful in describing varied phenomena and powerful algorithms with short descriptions. As Valiant points out: "the power [of mathematics and algorithms] comes from the implied generality, that knowledge of one equation alone will allow one to make accurate predictions about a host of situations not even conceived when the equation was first written down" (Valiant, 2013). Therefore, in order to obtain curiosity modules that can generalize over a very broad range of tasks and that are sophisticated enough to provide exploration guidance over very long horizons, we describe them in terms of general programs in a domain-specific language. Algorithms in this language will map a history of $(s_t, s_{t+1}, a_t, r_t)$ tuples into a proxy reward $\widehat{r}_t$.

Inspired by human-designed systems that compute and use intrinsic rewards, and to simplify the search, we decompose the curiosity module into two components: the first, $I$, outputs an intrinsic reward value $i_t$ based on the current experienced transition $(s_t, a_t, s_{t+1})$ (and past transitions $(s_{1:t-1}, a_{1:t-1})$ indirectly through its memory); the second, $\chi$, takes the current time-step $t$, the actual reward $r_t$, and the intrinsic reward $i_t$ (and, if it chooses to store them, their histories) and combines them to yield the proxy reward $\widehat{r}_t$. To ease generalization across different timescales, in practice, before feeding $t$ into $\chi$ we normalize it by the total length of the agent's lifetime, $T$.

Both programs consist of a directed acyclic graph (DAG) of modules with polymorphically typed inputs and outputs. As shown in figure 2, there are four classes of modules:

- **Input** modules (shown in blue), drawn from the set $\{s_t, a_t, s_{t+1}\}$ for the $I$ component and from the set $\{i_t, r_t\}$ for the $\chi$ component. They have no inputs, and their outputs have the type corresponding to the types of states and actions in whatever domain they are applied to, or the reals numbers for rewards.

- **Buffer and parameter** modules (shown in gray) of two kinds: FIFO queues that provide as output a finite list of the $k$ most recent inputs, and neural network weights initialized at random at the start of the program and which may (pink border) or may not (black border) get updated via back-propagation depending on the computation graph.

- **Functional** modules (shown in white), which compute output values given the inputs from their parent modules.

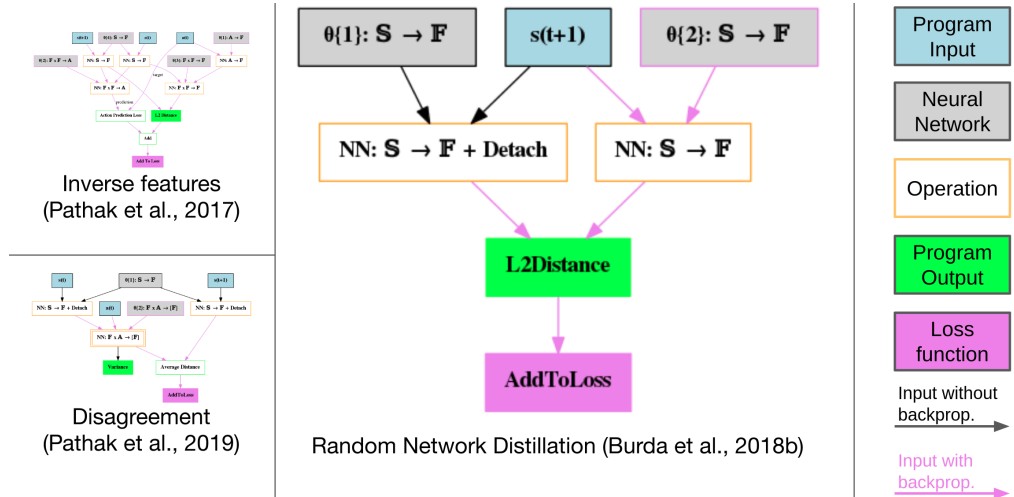

Figure 2: Example diagrams of published algorithms covered by our language (larger figures in the appendix). The green box represents the output of the intrinsic curiosity function, the pink box is the loss to be minimized. Pink arcs represent paths and networks along which gradients flow back from the minimizer to update parameters.

- **Update** modules (shown in pink), which are functional modules (such as k-Nearest-Neighbor) that either add variables to buffers or modules which add real-valued outputs to a global loss that will provide error signals for gradient descent.

A single node in the DAG is designated as the *output node* (shown in green): the output of this node is considered to be the output of the entire program, but it need not be a leaf node of the DAG.

On each call to a program (corresponding to one time-step of the system) the current input values and parameter values are propagated through the functional modules, and the output node's output is given to the RL algorithm. Before the call terminates, the FIFO buffers are updated and the adjustable parameters are updated via gradient descent using the Adam optimizer (Kingma & Ba, 2014). Most operations are differentiable and thus able to propagate gradients backwards. Some operations are not differentiable, including buffers (to avoid backpropagating through time) and "Detach" whose purpose is stopping the gradient from flowing back. In practice, we have multiple copies of the same agent running at the same time, with both a shared policy and shared curiosity module. Thus, we execute multiple reward predictions on a batch and then update on a batch.

Programs representing several published designs for curiosity modules that perform internal gradient descent, including inverse features (Pathak et al., 2017), random network distillation (RND) (Burda et al., 2018), and ensemble predictive variance (Pathak et al., 2019), are shown in figure 2 (bigger versions can be found in appendix A.3). We can also represent algorithms similar to novelty search (Lehman & Stanley, 2008) and $EX^2$ (Fu et al., 2017), which include buffers and nearest neighbor regression modules. Details on the data types and module library are given in appendix A.

A crucial, and possibly somewhat counter-intuitive, aspect of these programs is their use of neural network weight updates via gradient descent as a form of memory. In the parameter update step, all adjustable parameters are decremented by the gradient of the sum of the outputs of the loss modules, with respect to the parameters. This type of update allows the program to, for example, learn to make some types of predictions, online, and use the quality of those predictions in a state to modulate the proxy reward for visiting that state (as is done, for example, in RND).

Key to our program search are *polymorphic data types*: the inputs and outputs to each module are typed, but the instantiation of some types, and thus of some operations, depends on the environment. We have four types: reals $\mathbb{R}$, state space of the given environment $\mathbb{S}$, action space of the given environment $\mathbb{A}$ and feature space $\mathbb{F}$, used for intermediate computations and always set to $\mathbb{R}^{32}$ in our current implementation. For example, a neural network module going from $\mathbb{S}$ to $\mathbb{F}$ will be instantiated as a convolutional neural network if $\mathbb{S}$ is an image and as a fully connected neural network of the

appropriate dimension if $\mathbb{S}$ is a vector. Similarly, if we are measuring an error in action space $\mathbb{A}$ we use mean-squared error for continuous action spaces and negative log-likelihood for discrete action spaces. This facility means that the same curiosity program can be applied, independent of whether states are represented as images or vectors, or whether the actions are discrete or continuous, or the dimensionality of either.

This type of abstraction enables our meta-learning approach to discover curiosity modules that generalize *radically*, applying not just to new tasks, but to tasks with substantially different input and output spaces than the tasks they were trained on.

To clarify the semantics of these programs, we walk through the operation of the RND program in figure 2. Its only input is $s_{t+1}$, which might be an image or an input vector, which is processed by two NNs with parameters $\Theta_1$ and $\Theta_2$, respectively. The structure of the NNs (and, hence, the dimensions of the $\Theta_i$) depends on the type of $s_{t+1}$: if $s_{t+1}$ is an image, then they are CNNs, otherwise a fully connected networks. Each NN outputs a 32-dimensional vector; the $L_2$ distance between these vectors is the output of the program on this iteration, and is also the input to a loss module. So, given an input $s_{t+1}$, the output intrinsic reward is large if the two NNs generate different outputs and small otherwise. After each forward pass, the weights in $\Theta_2$ are updated to minimize the loss while $\Theta_1$ remains constant, which causes the trainable NN to mimic the output of the randomly initialized NN. As the program's ability to predict the output of the randomized NN on an input improves, the intrinsic reward for visiting that state decreases, driving the agent to visit new states.

To limit the search space and prioritize short, meaningful programs we limit the total number of modules of the computation graph to 7. Our language is expressive enough to describe many (but far from all) curiosity mechanisms in the existing literature, as well as many other potential alternatives, but the expressiveness leads to a very large search space. Additionally, removing or adding a single operation can drastically change the behavior of a program, making the objective function non-smooth and, therefore, the space hard to search. In the next section we explore strategies for speeding up the search over tens of thousands of programs.

## 3   IMPROVING THE EFFICIENCY OF OUR SEARCH

We wish to find curiosity programs that work effectively in a wide range of environments, from simple to complex. However, evaluating tens of thousands of programs in the most expensive environments would consume decades of GPU computation. Therefore, we designed multiple strategies for quickly discarding less promising programs and focusing computation on a few promising programs. In doing so, we take inspiration from efforts in the AutoML community (Hutter et al., 2018).

We divide these pruning efforts into three categories: simple tests that are independent of running the program in any environment, "filtering" by ruling out some programs based on poor performance in simple environments, and "meta-meta-RL": learning to predict which curiosity programs will produce good RL agents based on syntactic features.

### 3.1   PRUNING INVALID ALGORITHMS WITHOUT RUNNING THEM

Many programs are obviously bad curiosity programs. We have developed two heuristics to immediately prune these programs without an expensive evaluation.

- Checking that programs are not duplicates. Since our language is highly expressive, there are many non-obvious ways of getting equivalent programs. To find duplicates, we designed a randomized test where we identically seed two programs, feed them both identical fake environment data for tens of steps and check whether their outputs are identical.

- Checking that the loss functions cannot be minimized independently of the input data. Many programs optimize some loss depending on neural network regressors. If we treat inputs as uncontrollable variables and networks as having the ability to become any possible function, then for every variable, we can determine whether neural networks can be optimized to minimize it, independently of the input data. For example, if our loss function is $|NN_\theta(s)|^2$ the neural network can learn to make it $0$ by disregarding $s$ and optimizing the weights $\theta$ to $0$. We discard any program that has this property.

## 3.2 Pruning algorithms in cheap environments

Our ultimate goal is to find algorithms that perform well on many different environments, both simple and complex. We make two key observations. First, there may be only tens of reasonable programs that perform well on all environments but hundreds of thousands of programs that perform poorly. Second, there are some environments that are solvable in a few hundred steps while others require tens of millions. Therefore, a key idea in our search is to try many programs in cheap environments and only a few promising candidates in the most expensive environments. This was inspired by the effective use of sequential halving (Karnin et al., 2013) in hyper-parameter optimization (Jamieson & Talwalkar, 2016).

By pruning programs aggressively, we may be losing multiple programs that perform well on complex environments. However, by definition, these programs will tend to be less general and robust than those that succeed in all environments. Moreover, we seek generalization not only for its own sake, but also to ease the search since, even if we only cared about the most expensive environment, performing the complete search only in this environment would be impractical.

## 3.3 Predicting algorithm performance

Perhaps surprisingly, we find that we can predict program performance directly from program structure. Our search process bootstraps an initial training set of (program structure, program performance) pairs, then uses this training set to select the most promising next programs to evaluate. We encode each program's structure with features representing how many times each operation is used, thus having as many features as number of operations in our vocabulary. We use a $k$-nearest-neighbor regressor, with $k = 10$. We then try the most promising programs and update the regressor with their results. Finally, we add an $\epsilon$-greedy exploration policy to make sure we explore all the search space. Even though the correlation between predictions and actual values is only moderately high ($0.54$ on a holdout test), this is enough to discover most of the top programs searching only half of the program space, which is our ultimate goal. Results are shown in appendix C.

We can also prune algorithms during the training process of the RL agent. In particular, at any point during the meta-search, we use the top $K$ current best programs as benchmarks for all $T$ timesteps. Then, during the training of a new candidate program we compare its current performance at time $t$ with the performance at time $t$ of the top $K$ programs and stop the run if its performance is significantly lower. If the program is not pruned and reaches the final time-step $T$ with one of the top $K$ performances, it becomes part of the benchmark for the future programs.

## 4 Experiments

Our RL agent uses PPO (Schulman et al., 2017) based on the implementation by Kostrikov (2018) in PyTorch (Paszke et al., 2017). Our code (`https://github.com/mfranzs/meta-learning-curiosity-algorithms`) can take in any OpenAI gym environment (Brockman et al., 2016) with a specification of the desired exploration horizon $T$.

We evaluate each curiosity algorithm for multiple trials, using a seed dependent on the trial but independent of the algorithm, which leads to the PPO weights and curiosity data-structures being initialized identically on the same trials for all algorithms. As is common in PPO, we run multiple rollouts (5, except for MuJoCo which only has 1), with independent experiences but shared policy and curiosity modules. Curiosity predictions and updates are batched across these rollouts, but not across time. PPO policy updates are batched both across rollouts and multiple timesteps.

### 4.1 First search phase in simple environment

We start by searching for a good intrinsic curiosity program $I$ in a purely exploratory environment, designed by Chevalier-Boisvert et al. (2018), which is an image-based grid world where agents navigate in an image of a 2D room either by moving forward in the grid or rotating left or right. We optimize the total number of distinct cells visited across the agent's lifetime. This allows us to evaluate intrinsic reward programs in a fast and simple environment, without worrying about combining it with external reward.

To bias towards simple, interpretable algorithms and keep the search space manageable, we search for programs with at most 7 operations. We first discard duplicate and invalid programs, as described in section 3.1, resulting in about 52,000 programs. We then randomly split the programs across 4 machines, each with 8 Nvidia Tesla K80 GPUs for 10 hours; thus a total of 13 GPU days.

Each machine finds the highest-scoring 625 programs in its section of the search space and prunes programs whose partial learning curve is statistically significantly lower than the current top 625 programs. To do so, after every episode of every trial, we check whether $mean_{program}(step) \leq mean_{top625}(step) - 2std_{top625} - std_{program}$. Thus, we account for both inter-program variability among the top 625 programs and intra-program variability among multiple trials of the same program.

We use a 10-nearest-neighbor regressor to predict program performance and choose the next program to evaluate with an $\epsilon$-greedy strategy, choosing the best predicted program $90\%$ of the time and a random program $10\%$ of the time. By doing this, we try the most promising programs early in our search. This is important for two reasons: first, we only try 26,000 programs, half of the whole search space, which we estimated from earlier results (shown in figure 8 in the appendix) would be enough to get $88\%$ of the top $1\%$ of programs. Second, the earlier we run our best programs, the higher the bar for later programs, thus allowing us to prune them earlier, further saving computation time. Searching through this space took a total of 13 GPU days. As shown in figure 9 in the appendix, we find that most programs perform relatively poorly, with a long tail of programs that are statistically significantly better, comprising roughly $0.5\%$ of the whole program space.

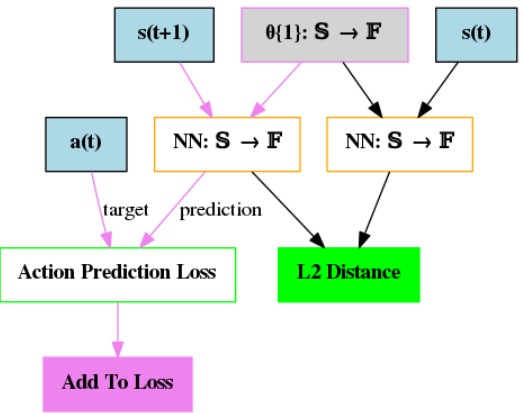

Figure 3: Fast Action-Space Transition(FAST): top-performing intrinsic curiosity algorithm discovered in our phase 1 search.

The highest scoring program (a few other programs have lower average performance but are statistically equivalent) is surprisingly simple and meaningful, comprised of only 5 operations, even though the limit was 7. This program, which we call FAST (Fast Action-Space Transition), is shown in figure 3; it trains a single neural network (a CNN or MLP depending on the type of state) to predict the action from $s_{t+1}$ and then compares its predictions based on $s_{t+1}$ with its predictions based on $s_t$, generating high intrinsic reward when the difference is large. The *action prediction loss* module either computes a softmax followed by NLL loss or appends zeros to the action to match dimensions and applies MSE loss, depending on the type of the action space. Note that this is not the same as rewarding taking a different action in the previous time-step. The network predicting the action is learning to imitate the policy learned by the internal RL agent, because the curiosity module does not have direct access to the RL agent's internal state.

Of the top 16 programs, 13 are variants of FAST, including versions that predict the action from $s_t$ instead of $s_{t+1}$. The other 3 are variants of a more complex program that is hard to understand at first glance, but we finally determined to be using ideas similar to cycle-consistency in the GAN literature Zhu et al. (2017) (we thus name it Cycle-consistency intrinsic motivation); the diagram and explanation are in figure 10 in the appendix. Interestingly, to the best of our knowledge neither algorithm had been proposed before: we conjecture the former was too simple for humans to believe it would be effective and the latter too hard for humans to design, as it was already very hard to understand in hindsight.

## 4.2 TRANSFERRING TO NEW ENVIRONMENTS

Our reward combiner was developed in *lunar lander* (the simplest environment with meaningful extrinsic reward) based on the best program among a preliminary set of 16,000 programs (which resembled Random Network Distillation; its computation graph is shown in appendix E). Among a set

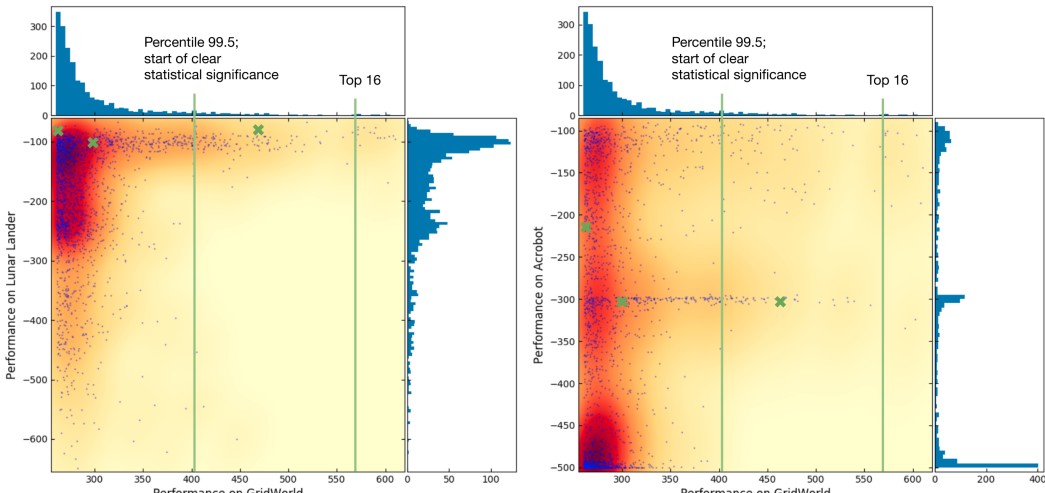

Figure 4: Correlation between program performance in gridworld and in harder environments (lunar lander on the left, acrobot on the right), using the top 2,000 programs in gridworld. Performance is evaluated using mean reward across *all* learning episodes, averaged over trials (two trials for acrobot / lunar lander and five for gridworld). The high number of algorithms performing around -300 in the middle of the right plot is an artifact of averaging the performance of two seeds and the mean performance in Acrobot having two peaks. Almost all intrinsic curiosity programs that had statistically significant performance for grid world also do well on the other two environments. In green, the performance of three published works; in increasing gridworld performance: disagreement (Pathak et al., 2019), inverse features (Pathak et al., 2017) and random distillation (Burda et al., 2018).

of 2,500 candidates (with 5 or fewer operations) the best reward combiner discovered by our search was $\widehat{r}_t = \frac{(1+i_t-t/T)\cdot i_t+t/T\cdot r_t}{1+i_t}$. Notice that for $0 < i_t \ll 1$ (usually the case) this is approximately $\widehat{r}_t \approx i_t^2 + (1-t/T)i_t + (t/T)r_t$, which is a down-scaled version of intrinsic reward plus a linear interpolation that ranges from all intrinsic reward at $t = 0$ to all extrinsic reward at $t = T$. In future work, we hope to co-adapt the search for intrinsic reward programs and combiners as well as find multiple reward combiners.

Given the fixed reward combiner and the list of 2,000 selected programs found in the image-based grid world, we evaluate the programs on both *lunar lander* and *acrobot*, in their discrete action space versions. Notice that both environments have much longer horizons than the image-based grid world (37,500 and 50,000 vs 2,500) and they have vector-based, rather than image-based, inputs. The results in figure 4 show good correlation between performance on grid world and on each of the new environments. Especially interesting is that, for both environments, when intrinsic reward in grid world is above 400 (the lowest score that is statistically significantly good), performance on the other two environments is also good in more than $90\%$ of cases.

Finally, we evaluate on two MuJoCo environments (Todorov et al., 2012): *hopper* and *ant*. These environments have more than an order of magnitude longer exploration horizon than acrobot and lunar lander, exploring for 500K time-steps, as well as continuous action-spaces instead of discrete. We then compare the best 16 programs on grid world (most of which also did well on lunar lander and acrobot) to four weak baselines (constant 0,-1,1 intrinsic reward and Gaussian noise reward) and three published algorithms expressible in our language (shown in figure 2). We run two trials for each algorithm and pool all results in each category to get a confidence interval for the mean of that category. All trials used the reward combiner found on lunar lander. For both environments we find that the performance of our top programs is statistically equivalent to published work and significantly better than the weak baselines, confirming that we meta-learned good curiosity programs.

Note that we meta-trained our intrinsic curiosity programs only on one environment (GridWorld) and showed they generalized well to other very different environments: they perform better than published works in this meta-train task and one meta-test task (Acrobot) and on par in the other 3 tasks meta-test tasks. Adding more meta-training tasks would be as simple as standardising the perfor-

| Class | Ant | Hopper |
|---|---|---|
| Baseline algorithms | [-95.3, -39.9] | [318.5, 525.0] |
| Meta-learned algorithms | [+67.5, +80.0] | [589.2, 650.6] |
| Published algorithms | [+67.4, +98.8] | [627.7, 692.6] |

Table 1: Meta-learned algorithms perform significantly better than constant rewards and statistically equivalently to published algorithms found by human researchers (see 2). The table shows the confidence interval (one standard deviation) for the mean performance (across trials, across algorithms) for each algorithm category. Performance is defined as mean episode reward for all episodes.

mance within each task (to make results comparable) and then selecting the programs with best mean performance. We chose to only meta-train on a single, simple, task because it (surprisingly!) already gave great results, highlighting the broad generalization of meta-learning program representations.

## 5 RELATED WORK

In some regards our work is similar to neural architecture search (NAS) (Stanley & Miikkulainen, 2002; Zoph & Le, 2016; Elsken et al., 2018; Pham et al., 2018) or hyperparameter optimization for deep networks (Mendoza et al., 2016), which aim at finding the best neural network architecture and hyper-parameters for a particular task. However, in contrast to most (but not all, see Zoph et al. (2018)) NAS work, we want to generalize to many environments instead of just one. Moreover, we search over programs, which include non-neural operations and data structures, rather than just neural-network architectures, and decide what loss functions to use for training. Our work also resembles work in the AutoML community (Hutter et al., 2018) that searches in a space of programs, for example in the case of SAT solving (KhudaBukhsh et al., 2009) or auto-sklearn (Feurer et al., 2015) and concurrent work on learning loss functions to replace cross-entropy for training a fixed architecture on MNIST and CIFAR (Gonzalez & Miikkulainen, 2019; 2020). Although we took inspiration from ideas in that community (Jamieson & Talwalkar, 2016; Li et al., 2016), our algorithms specify both how to compute their outputs and their own optimization objectives in order to work well in synchrony with an expensive deep RL algorithm.

There has been work on meta-learning with genetic programming (Schmidhuber, 1987), searching over mathematical operations within neural networks (Ramachandran et al., 2017; Gaier & Ha, 2019), searching over programs to solve games (Wilson et al., 2018; Kelly & Heywood, 2017; Silver et al., 2019) and to optimize neural networks (Bengio et al., 1995; Bello et al., 2017), and neural networks that learn programs (Reed & De Freitas, 2015; Pierrot et al., 2019). Our work uses neural networks as basic operations within larger algorithms. Finally, modular meta-learning (Alet et al., 2018; 2019) trains the weights of small neural modules and transfers to new tasks by searching for a good composition of modules; as such, it can be seen as a (restricted) dual of our approach.

There has been much interesting work in designing intrinsic curiosity algorithms. We take inspiration from many of them to design our domain-specific language. In particular, we rely on the idea of using neural network training as an implicit memory, which scales well to millions of time-steps, as well as buffers and nearest-neighbour regressors. As we showed in figure 2 we can represent several prominent curiosity algorithms. We can also generate meaningful algorithms similar to novelty search (Lehman & Stanley, 2008) and $EX^2$ (Fu et al., 2017); which include buffers and nearest neighbours. However, there are many exploration algorithm classes that we do not cover, such as those focusing on generating goals (Srivastava et al., 2013; Kulkarni et al., 2016; Florensa et al., 2018), learning progress (Oudeyer et al., 2007; Schmidhuber, 2008; Azar et al., 2019), generating diverse skills (Eysenbach et al., 2018), stochastic neural networks (Florensa et al., 2017; Fortunato et al., 2017), count-based exploration (Tang et al., 2017) or object-based curiosity measures (Forestier & Oudeyer, 2016). Finally, part of our motivation stems from Taïga et al. (2019) showing that some bonus-based curiosity algorithms have trouble generalising to new environments.

There have been research efforts on meta-learning exploration policies: Duan et al. (2016); Wang et al. (2017) learn an LSTM that explores an environment for one episode, retains its hidden state and is spawned in a second episode in the same environment; by training the network to maximize the reward in the second episode alone it learns to explore efficiently in the first episode. Stadie et al. (2018) improves their exploration and that of Finn et al. (2017) by considering the importance of

sampling in RL policies. Gupta et al. (2018) combine gradient-based meta-learning with a learned latent exploration space in which they add structured noise for meaningful exploration. Closer to our formulation, Zheng et al. (2018) parametrize an intrinsic reward function which influences policy-gradient updates in a differentiable manner, allowing them to backpropagate through a single step of the policy-gradient update to optimize the intrinsic reward function for a single task. In contrast to all three of these methods, we search over algorithms, which will allows us to generalize more broadly and to consider the effect of exploration on up to $10^5 - 10^6$ time-steps instead of the $10^2 - 10^3$ of previous work. Finally, Chiang et al. (2019); Faust et al. (2019) have a setting similar to ours where they modify reward functions over the entire agent's lifetime, but instead of searching over intrinsic curiosity algorithms they tune the parameters of a hand-designed reward function.

Related work on meta-learning (Schmidhuber, 1987; Thrun & Pratt, 1998; Clune, 2019) and efforts to increase its generalization can be found in appendix B. Closest to our work, evolved policy gradients (EPG, Houthooft et al. (2018)) use evolutionary strategies to meta-learn a neural network that acts as a loss function and is used to train a policy network. EPG generalizes by meta-training with target locations east of the start location and meta-testing with target locations to the west. In contrast, we showed that by meta-learning programs, we can generalize between radically different environments, not just goal variations of a single environment. Concurrent to our work, Kirsch et al. (2019) also show generalization capabilities between environments similar to ours (lunar lander, hopper and half-cheetah). Their approach transfers a parametric representation, for which it is unclear how to adapt the learned neural losses to an unseen environment with a different observation space. Their approach thus does not encode states into the loss function, which is critical for efficient exploration. In contrast, our algorithms can leverage polymorphic data types that adapt the neural networks to the environment they are running in, adapting both the size and the type of network (CNN vs MLP) running in each environment.

## 6 CONCLUSIONS

In this work, we proposed to meta-learn algorithms and show that by transferring programs we can generalize between tasks much more varied than previously possible in meta-RL, even between those with different input or output spaces. In many settings, however, the input and output space remain the same as we change tasks. This opens the possibility of getting the best of both worlds by meta-learning weights along with structure, thus simultaneously transferring domain-specific knowledge in the weights and higher-level algorithmic knowledge in the architecture. In addition, we note that the approach of meta-learning programs instead of network weights may have further applications beyond finding curiosity algorithms, such as meta-learning optimization algorithms or even meta-learning meta-learning algorithms. Our relatively modest compute (2 GPU-weeks) and a simple search method restricted us to a medium-sized search space, but we expect that future work could search over significantly bigger spaces. It thus may be possible to automatically search for new machine learning algorithms from more fundamental building blocks for a wide variety of problems.

### ACKNOWLEDGMENTS

We thank Kelsey Allen, Peter Karkus, Kevin Smith, Josh Tenenbaum and the rest of the Honda-CMM MIT team for their insightful feedback. We thank Chris Lu for his idea on what the algorithm in figure 10 is computing. We also want to thank Bernadette Bucher, Chelsea Finn, Abhishek Gupta, Deepak Pathak, Lerrel Pinto, Oleh Rybkin, Karl Schmeckpeper and Joaquin Vanschoren for valuable conversations. Finally, we also want to thank Maria Bauza and Tej Chajed for their feedback on early drafts and Clement Gehring for his help setting up the experiments.

We gratefully acknowledge support from NSF grants 1523767 and 1723381, AFOSR grant FA9550-17-1-0165, ONR grant N00014-18-1-2847, the Honda Research Institute, SUTD Temasek Laboratories and the MIT Quest for Intelligence. Any opinions, findings, and conclusions or recommendations expressed in this material do not necessarily reflect the views of our sponsors.

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

# A    DETAILS OF OUR DOMAIN-SPECIFIC LANGUAGE FOR CURIOSITY ALGORITHMS

We have the following types. Note that $\mathbb{S}$ and $\mathbb{A}$ get defined differently for every environment.

- $\mathbb{R}$: real numbers such as $r_t$ or the dot-product between two vectors.
- $\mathbb{R}^+$: numbers guaranteed to be positive, such as the distance between two vectors. The only difference to our program search between $\mathbb{R}$ and $\mathbb{R}^+$ is in pruning programs that can optimize objectives without looking at the data. For $\mathbb{R}^+$ we check whether they can optimize down to 0, for $\mathbb{R}$ we check whether they can optimize to arbitrarily negative values.
- state space $\mathbb{S}$: the environment state, such as a matrix of pixels or a vector with robot joint values. The particular form of this type is adapted to each environment.
- action space $\mathbb{A}$: either a 1-hot description of the action or the action itself. The particular form of this type is adapted to each environment.
- feature-space $\mathbb{F} = \mathbb{R}^{32}$: a space mostly useful to work with neural network embeddings. For simplicity, we only have a single feature space.
- List[$\mathbb{X}$]: for each type we may also have a list of elements of that type. All operations that take a particular type as input can also be applied to lists of elements of that type by mapping the function to every element in the list. Lists also support extra operations such as average or variance.

## A.1    CURIOSITY OPERATIONS

| Operation | Input type(s) | State | Output type |
|---|---|---|---|
| Add | $\mathbb{R}, \mathbb{R}$ | | $\mathbb{R}$ |
| RunningNorm | $\mathbb{R}$ | $\mathbb{R}$ | $\mathbb{R}$ |
| VariableAsBuffer | $\mathbb{X}$ | $List[\mathbb{X}]$ | $List[\mathbb{X}]$ |
| NearestNeighborRegressor | $\mathbb{F}, \mathbb{F}$ | $List[\mathbb{F}]$ | $\mathbb{F}$ |
| SubtractOneTenth | $\mathbb{R}$ | | $\mathbb{R}$ |
| NormalDistribution | | | $\mathbb{R}$ |
| Subtract | $\mathbb{R}, \mathbb{R}$ | | $\mathbb{R}$ |
| Sqrt(Abs(x)) | $\mathbb{R}$ | | $\mathbb{R}^+$ |
| NN $\mathbb{F}, \mathbb{F} \to \mathbb{F}$ | $\mathbb{F}, \mathbb{F}$ | $\Theta_{\mathbb{F}, \mathbb{F} \to \mathbb{F}}$ | $\mathbb{F}$ |
| NN $\mathbb{F}, \mathbb{F} \to \mathbb{A}$ | $\mathbb{F}, \mathbb{F}$ | $\Theta_{\mathbb{F}, \mathbb{F} \to \mathbb{A}}$ | $\mathbb{A}$ |
| NN $\mathbb{F} \to \mathbb{A}$ | $\mathbb{F}$ | $\Theta_{\mathbb{F} \to \mathbb{A}}$ | $\mathbb{A}$ |
| NN $\mathbb{A} \to \mathbb{F}$ | $\mathbb{A}$ | $\Theta_{\mathbb{A} \to \mathbb{F}}$ | $\mathbb{F}$ |
| (C)NN | $\mathbb{S}$ | $\Theta_{\mathbb{S} \to \mathbb{F}}$ | $\mathbb{F}$ |
| (C)NN, Detach | $\mathbb{S}$ | $\Theta_{\mathbb{S} \to \mathbb{F}}$ | $\mathbb{F}$ |
| (C)NNEnsemble | $\mathbb{S}$ | $5x\Theta_{\mathbb{S} \to \mathbb{F}}$ | $List[\mathbb{F}]$ |
| NN Ensemble $\mathbb{F} \to \mathbb{F}$ | $\mathbb{F}$ | $5x\Theta_{\mathbb{F} \to \mathbb{F}}$ | $List[\mathbb{F}]$ |
| NN Ensemble $\mathbb{F}, \mathbb{F} \to \mathbb{F}$ | $\mathbb{F}, \mathbb{F}$ | $5x\Theta_{\mathbb{F}, \mathbb{F} \to \mathbb{F}}$ | $List[\mathbb{F}]$ |
| NN Ensemble $\mathbb{F}, \mathbb{A} \to \mathbb{F}$ | $\mathbb{F}, \mathbb{A}$ | $5x\Theta_{\mathbb{A}, \mathbb{F} \to \mathbb{F}}$ | $List[\mathbb{F}]$ |
| MinimizeValue | $\mathbb{R}$ | Adam | |
| L2Norm | $\mathbb{X}$ | | $\mathbb{R}^+$ |
| L2Distance | $\mathbb{X}, \mathbb{X}$ | | $\mathbb{R}$ |
| ActionSpaceLoss | $\mathbb{X}, \mathbb{A}$ | | $\mathbb{R}^+$ |
| DotProduct | $\mathbb{X}, \mathbb{X}$ | | $\mathbb{R}$ |
| Add | $\mathbb{X}, \mathbb{X}$ | | $\mathbb{X}$ |
| Detach | $\mathbb{X}$ | | $\mathbb{X}$ |
| Mean | $List[\mathbb{R}]$ | | $\mathbb{R}$ |
| Variance | $List[\mathbb{X}]$ | | $\mathbb{R}^+$ |
| Mean | $List[\mathbb{X}]$ | | $\mathbb{X}$ |
| Mapped L2 Norm | $List[\mathbb{X}]$ | | $List[\mathbb{R}]$ |
| Average Distance | $List[\mathbb{X}], \mathbb{X}$ | | $\mathbb{R}$ |
| Minus | $List[\mathbb{X}], \mathbb{X}$ | | $List[\mathbb{X}]$ |

Note that $\mathbb{X}$ stands for the option of being $\mathbb{F}$ or $\mathbb{A}$. NearestNeighborRegressor takes a query and a target, automatically creates a buffer of the target (thus keeps a list as a state) and answers based on the buffer. RunningNorm keeps track of the variance of the input and normalizes by that variance.

## A.2 REWARD COMBINER OPERATIONS

| Operation | Input type(s) | State | Output type |
|---|---|---|---|
| Constant $\{0.01,0.1,0.5,1\}$ | | | $\mathbb{R}$ |
| NormalDistribution | | | $\mathbb{R}$ |
| Add | $\mathbb{R}, \mathbb{R}$ | | $\mathbb{R}$ |
| Max | $\mathbb{R}, \mathbb{R}$ | | $\mathbb{R}$ |
| Min | $\mathbb{R}, \mathbb{R}$ | | $\mathbb{R}$ |
| WeightedNormalizedSum | $\mathbb{R}, \mathbb{R}, \mathbb{R}, \mathbb{R}$ | | $\mathbb{R}$ |
| RunningNorm | $\mathbb{R}$ | $\mathbb{R}$ | $\mathbb{R}$ |
| VariableAsBuffer | $\mathbb{R}$ | $List[\mathbb{R}]$ | $List[\mathbb{R}]$ |
| Subtract | $\mathbb{R}, \mathbb{R}$ | | $\mathbb{R}$ |
| Multiply | $\mathbb{R}, \mathbb{R}$ | | $\mathbb{R}$ |
| Sqrt(Abs(x)) | $\mathbb{R}$ | | $\mathbb{R}^+$ |
| Mean | $List[\mathbb{R}]$ | | $\mathbb{R}$ |

Note that $WeightedNormalizedSum(a, b, c, d) = \frac{ab+cd}{|a|+|c|}$. RunningNorm keeps track of the variance of the input and normalizes by that variance.

## A.3 TWO OTHER PUBLISHED ALGORITHMS COVERED BY OUR DSL

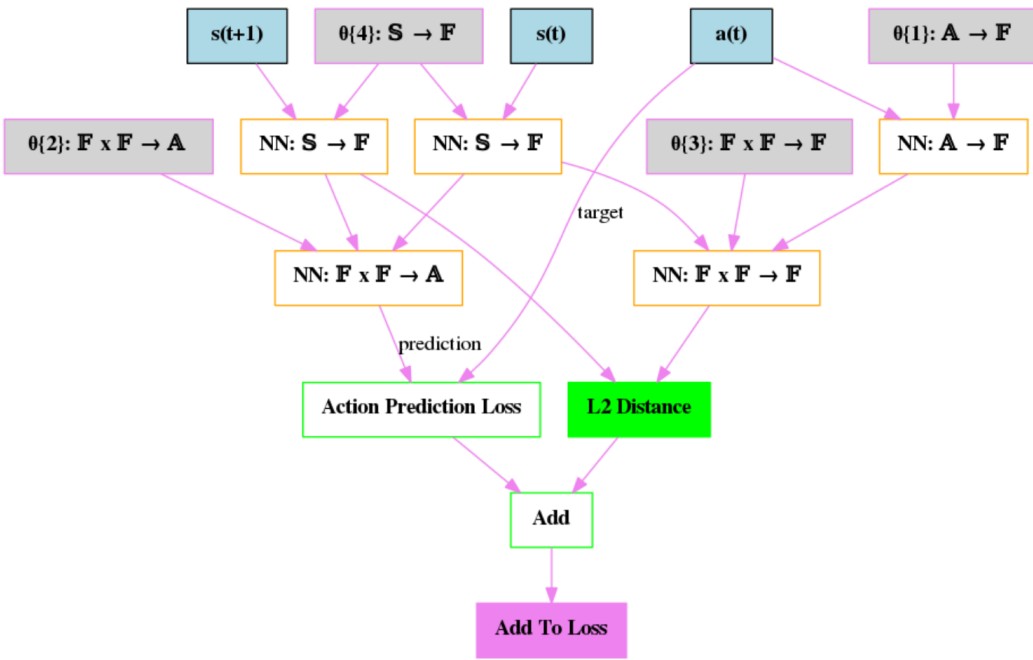

Figure 5: Curiosity by predictive error on inverse features by Pathak et al. (2017). In pink, paths and networks where gradients flow back from the minimizer.

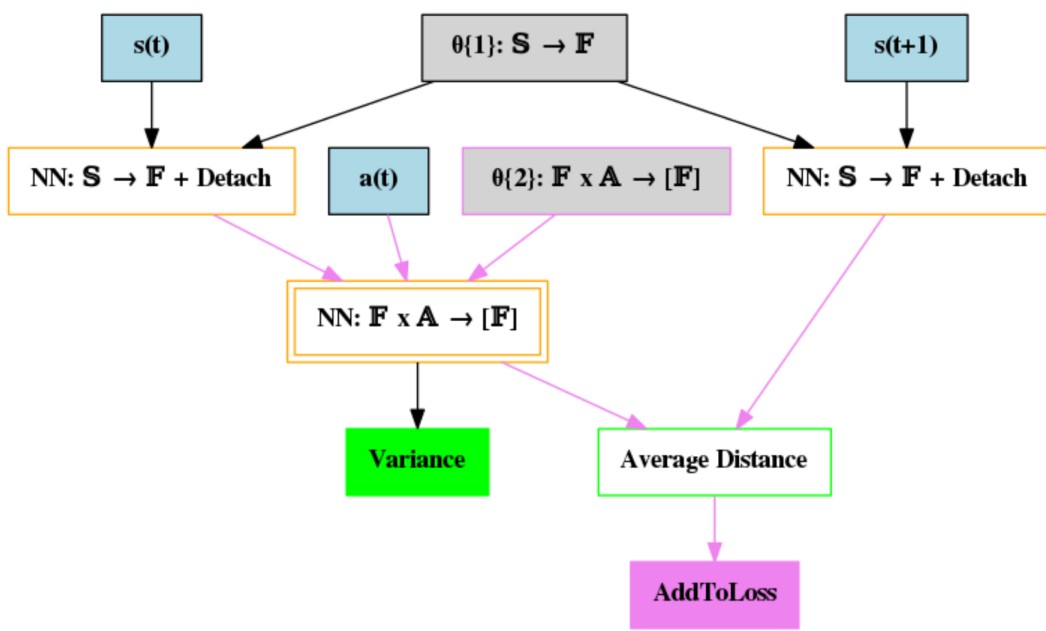

Figure 6: Curiosity by ensemble predictive variance Pathak et al. (2019). In pink, paths and networks where gradients flow back from the minimizer.

## B   RELATED WORK ON META-RL AND GENERALIZATION

Most work on meta-RL has focused on learning transferable feature representations or parameter values for quickly adapting to new tasks (Finn et al., 2017; Finn, 2018; Clavera et al., 2019) or improving performance on a single task (Xu et al., 2018; Veeriah et al., 2019). However, the range of variability between tasks is typically limited to variations of the same goal (such as moving at different speeds or to different locations) or generalizing to different environment variations (such as different mazes or different terrain slopes). There have been some attempts to broaden the spectrum of generalization, showing transfer between Atari games thanks to modularity (Fernando et al., 2017; Rusu et al., 2016) or proper pretraining (Parisotto et al., 2015). However, as noted by Nichol et al. (2018), Atari games are too different to get big gains with current feature-transfer methods; they instead suggest using different levels of the game *Sonic* to benchmark generalization. Moreover, Yu et al. (2019) recently proposed a benchmark of many tasks. Wang et al. (2019) automatically generate different terrains for a bipedal walker and transfer policies between terrains, showing that this is more effective than learning a policy on hard terrains from scratch; similar to our suggestion in section 3.2. In contrast to these methods, we aim at generalization between completely different environments, even between environments that do not share the same state and action spaces.

## C  PREDICTING ALGORITHM PERFORMANCE

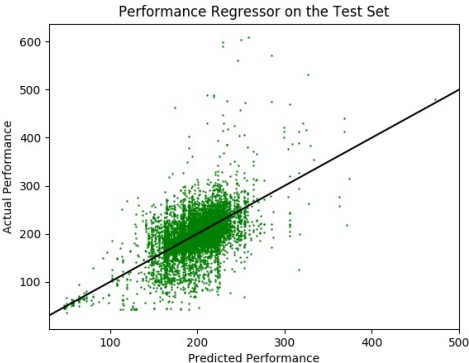

Figure 7: Predicting algorithm performance from the structure of the program alone. Comparison between predicted and actual performance on a test set; showing a correlation of $0.54$. In black, the identity line.

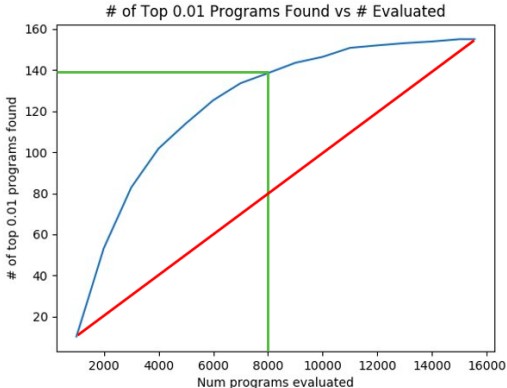

Figure 8: Predicting algorithm performance allows us to find the best programs faster. We investigate the number of the top 1% of programs found vs. the number of programs evaluated, and observe that the optimized search (in blue) finds 88% of the best programs after only evaluating 50% of the programs (highlighted in green). The naive search order would have only found 50% of the best programs at that point.

# D   PERFORMANCE ON GRID WORLD

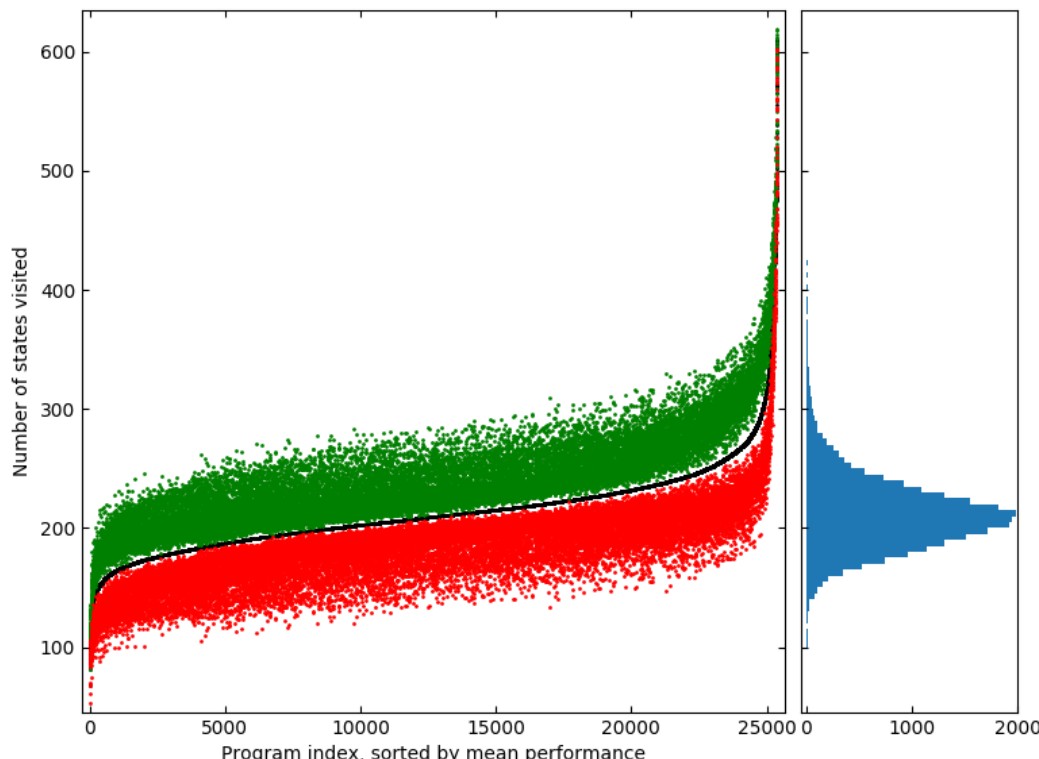

Figure 9: In black, mean performance across 5 trials for all 26,000 programs evaluated (out of their finished trials). In green mean plus one standard deviation for the mean estimate and in red one minus one standard deviation for the mean estimate. On the right, you can see program means form roughly a gaussian distribution of very big noise (thus probably not significant) with a very small (between $0.5\%$ and $1\%$ of programs) long tail of programs with statistically significantly good performance (their red dots are much higher than almost all green dots), composed of algorithms leading to good exploration.

## E INTERESTING PROGRAMS FOUND BY OUR SEARCH

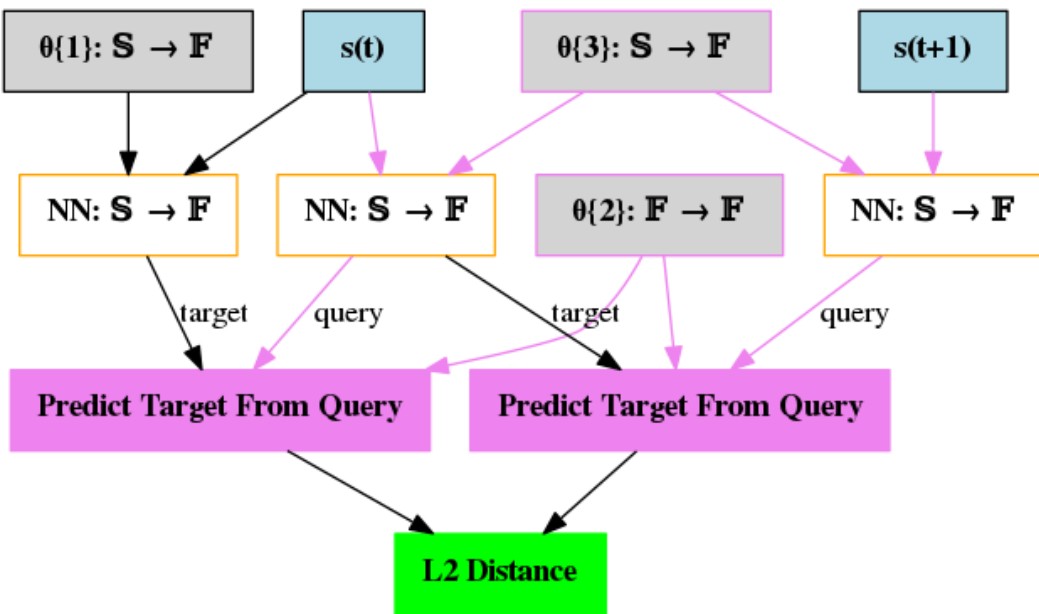

Figure 10: *Cycle-Consistency* Intrinsic Motivation algorithm, found by our search (3 of the top 16 programs on grid world are variants of this program). The purple *Predict Target From Query* boxes feed the query to a neural network, return the prediction as output and add the prediction loss to the optimization, back-propagating to the network and the query, but not the target. Notice that $\theta_1$ is not getting trained because no loss back-propagates there; thus producing a random feature embedding $s_f(t)$ from $s(t)$. The algorithm combines several concepts seen in the literature, such as an untrained network like RND Burda et al. (2018) and predicting another state in feature space like Pathak et al. (2017; 2019), but also includes weight sharing between both predictions, which makes the algorithm hard to interpret at first sight, see below for an in-depth explanation.

One can give meaning to the role of all 3 neural networks by considering how they contribute to minimizing the loss. To do so, let us name the networks: $\theta\{1\}$ (as labeled in the figure) as $r_{\theta_1}$ (for random embedding), $\theta\{2\}$ as $b_{\theta_2}$ (for backwards) and $\theta\{3\}$ as $fr_{\theta_3}$ (for forward and random embedding) and look at the algorithm in equation form:

$$i_t = \|b_{\theta_2}\left(fr_{\theta_3}(s_t)\right) - b_{\theta_2}\left(fr_{\theta_3}(s_{t+1})\right)\|$$
$$\theta_1 \text{ is kept at its random initialization}$$
$$\theta_2 := \theta_2 - \eta\frac{\partial}{\partial\theta_2}\Big(\|b_{\theta_2}\left(fr_{\theta_3}(s_t)\right) - r_{\theta_1}(s_t)\| +$$
$$\|b_{\theta_2}\left(fr_{\theta_3}(s_{t+1})\right) - fr_{\theta_3}(s_t)\|\Big)$$
$$\theta_3 := \theta_3 - \eta\frac{\partial}{\partial\theta_3}\Big(\|b_{\theta_2}\left(fr_{\theta_3}(s_t)\right) - r_{\theta_1}(s_t)\|\Big) \tag{1}$$

We can see that $r_{\theta_1}$ will indeed be a random embedding because the network is randomly initialized and is not trained. Then, we observe that the second term in the loss for $\theta_2$, which does not involve $\theta_3$ and thus $\theta_2$ has to minimize alone, is $\|b_{\theta_2}\left(fr_{\theta_3}(s_{t+1})\right) - fr_{\theta_3}(s_t)\|$. In this term, $b_{\theta_2}$ receives a transformation of $s_{t+1}$ and has to make it very similar to the same transformation applied to $s_t$; therefore, this term is similar to cycle-consistency found in some other parts of machine learning Zhu et al. (2017) and $b_{\theta_2}$ must act like a backward model. Finally, looking at the minimization of $\theta_3$ receives the original $s_t$ and has to output a vector such that the backward model will bring it close to

the random embedding of $s_t$. Therefore $\theta_3$ must learn a forward model composed with the random embedding of $\theta_1$. Finally, we see that the algorithm outputs $\|b_{\theta_2}\left(fr_{\theta_3}(s_t)\right) - b_{\theta_2}\left(fr_{\theta_3}\left(s_{t+1}\right)\right)\|$, going forward and backward for both $s_{t+1}$ and $s_t$ and comparing the difference. In summary, this distance combines errors in the cycle-consistency of predictions (which will be higher in unvisited parts of the state) with distance in the random embedding space between $s(t)$ and $s(t+1)$, i.e. moving to a very different state.

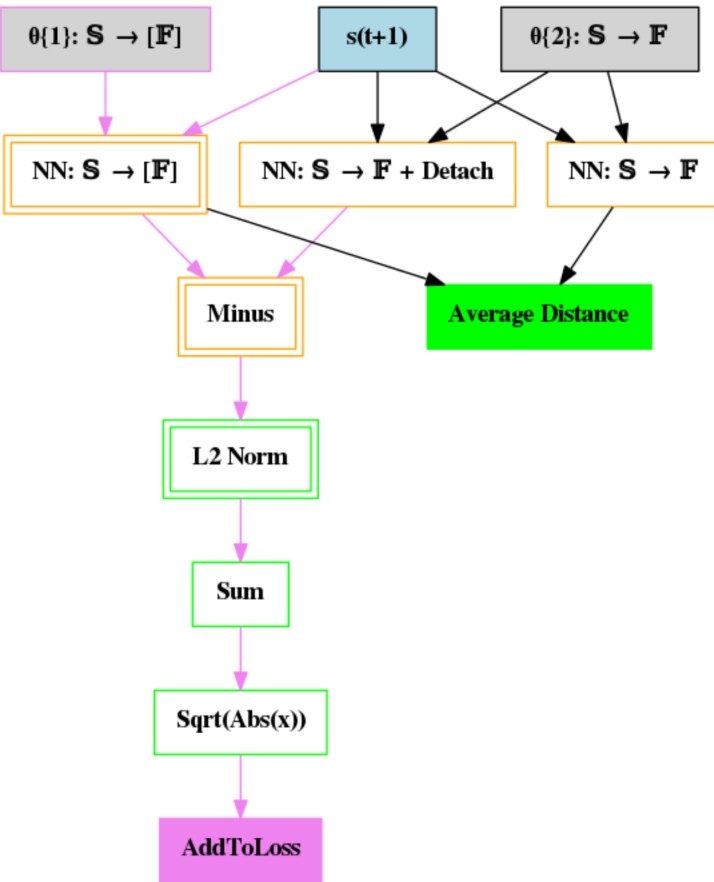

Figure 11: Top variant in preliminary search on grid world; variant on random network distillation using an ensemble of trained networks instead of a single one.

