# OpenReview forum: "Meta-learning curiosity algorithms"
_ICLR.cc/2020/Conference — Accept (Poster)_

### Official Review · AnonReviewer1 · 2019-10-22
**Official Blind Review #1**

**Rating:** 6

**Review:**

Instead of hand-designing exploration bonuses and intrinsic reward, the paper proposes to view the curiosity algorithms as programs described with domain-specific language (DSL), then search programs which allows RL agents to optimize the environment reward combined with the curiosity reward generated by the program. The search produces programs similar to algorithms proposed in literature and also some strategies which generalize well. In order to make the search tractable, the authors develop a few criteria: 1) evaluate the program on relatively simple and short-horizon domains; 2) predict the performance of the program and rank them; 3) stop the agents if the learning curves do not look good after long enough of training steps.

This is a very interesting idea, and it's partially inspired by the architecture search line of research. It would be great if the authors could provide more information about "predicting algorithm performance" section.

I find it very interesting and exciting to see programs like Figure 3. They are quite interpretable. The results on Table 1 are really exciting, they show that the searched programs could be generalized into other unseen tasks.

**Experience Assessment:**

I have published one or two papers in this area.

**Review Assessment: Checking Correctness Of Derivations And Theory:**

I assessed the sensibility of the derivations and theory.

**Review Assessment: Checking Correctness Of Experiments:**

I assessed the sensibility of the experiments.

**Review Assessment: Thoroughness In Paper Reading:**

I made a quick assessment of this paper.

---

> ### Author Response · Authors · 2019-11-15
> **Thanks for your helpful review!**
>
> With respect to evidence for predicting performance directly from program structure, we had a plot in appendix C showing we could find 88% of top programs after searching through half the program space, but we did not point to it within section 3.3. We have now extended the details of section 3.3, properly made a pointer to appendix C and added a second plot to the appendix showing the correlation between predicted performance and actual performance.
>
> Finally, we have added a general answer with comments we find relevant to all three reviewers.

---

### Official Review · AnonReviewer3 · 2019-10-23
**Official Blind Review #3**

**Rating:** 6

**Review:**

This paper presents an algorithm to generate curiosity modules for reinforcement learning. The authors define a program language which can represent many possible curiosity modules that include training neural networks, replay buffers, etc. It also presents an approach to searching for the best curiosity module in this set, which is some various ways to do pruning and to determine which methods to try.

The paper is very novel - the idea of developing a domain specific language full of building blocks to represent various curiosity modules is unique and interesting.

The search over curiosity modules is a bit mis-represented I think. In the introduction, it gives the impression that part of the algorithm is to search over these curiosity modules, and also that it's to find the best one that works across a wide set of tasks. Instead the search method is a separate procedure outside of the algorithm and most of the search steps are performed on individual tasks instead of over a set of tasks.

In Sec 3.3, you say that "perhaps surprisingly, we find that we can predict performance directly from program structure," but you never provide any evidence of doing so.

The simple environment that you used is a bit contrived, rather than taking a normal task, the goal itself is to do complete exploration (maximize the total number of pixels visited). It seems like the combination of the intrinsic curiosity program here with the reward combiner is that the intrinsic curiosity program should be only about complete exploration, and then the combiner is responsible for balancing that with task rewards. You should be more explicit that in this first part of the search you're only looking at the intrinsic curiosity program, without the combiner, and therefore do not want a task with extrinsic rewards. This breakdown of searching for the intrinsic curiosity program first and the combiner later seems like another important aspect of making your search efficient.

The main drawback of this paper is that there are little comparisons to related work. The only methods compared to are ones where the curiosity method is expressible in the language of the method.



**Experience Assessment:**

I have published one or two papers in this area.

**Review Assessment: Checking Correctness Of Derivations And Theory:**

N/A

**Review Assessment: Checking Correctness Of Experiments:**

I carefully checked the experiments.

**Review Assessment: Thoroughness In Paper Reading:**

I read the paper at least twice and used my best judgement in assessing the paper.

---

> ### Author Response · Authors · 2019-11-15
> **Thanks for your helpful review!**
>
> With respect to evidence for predicting performance directly from program structure, we had a plot in appendix C, but forgot to point to it within section 3.3. We have now extended the details of that section, properly made a pointer to appendix C and added a second plot to the appendix with further details.
>
> As the reviewer mentions, our framework describes searching curiosity algorithms that do well on multiple tasks, but in the experiment section we only search in one task. We note, however, that this analysis could easily be performed by a simple extension; for example, we could combine the data from the “Gridworld vs. Lunar Lander” plot in Figure 4, normalize the performance in each environment, and return algorithms sorted by their mean standardized performance. We chose to keep 1 meta-training task because we think the fact that one can transfer only from a single, simple, unrelated task is a stronger message (and a result we did not initially expect!). We have added a paragraph at the end of section 4 to clarify this point.
>
> Although it is true that our baselines can be expressed in our language, we note that they are widely regarded as very strong algorithms within the curiosity literature, which then caused us to design the language around them and use them as the strongest benchmark we could think of. Another alternative, which we deemed to be much weaker, would have been comparing to previous meta-learning algorithms (none of which have been shown to be capable of transferring to radically different environments). We add more details about this choice in a separate answer for all three reviewers.

---

### Official Review · AnonReviewer2 · 2019-10-23
**Official Blind Review #2**

**Rating:** 6

**Review:**

This paper proposes to meta-learn a curiosity module via neural architecture search. The curiosity module, which outputs a meta-reward derived from the agent’s history of transitions, is optimized via black box search in order to optimize the agent’s lifetime reward over a (very) long horizon. The agent in contrast is trained to maximize the episodic meta-reward and acts greedily wrt. this intrinsic reward function. Optimization of the curiosity module takes the form of an epsilon-greedy search, guided by a nearest-neighbor regressor which learns to predict the performance of a given curiosity program based on hand-crafted program features. The program space itself composes standard building blocks such as neural networks, non-differentiable memory modules, nearest neighbor regresses, losses, etc. The method is evaluated by learning a curiosity module on the MiniGrid environment (with the true reward being linked to discovering new states in the environment) and evaluating it on Lunar Lander and Acrobot. A reward combination module (which combines intrinsic and extrinsic rewards) is further evaluated on continuous-control tasks (Ant, Hopper) after having been meta-trained on Lunar Lander. The resulting agents are shown to match the performance of some recent published work based on curiosity and outperforms simple baselines.

This is an interesting, clear and well written paper which covers an important area of research, namely how to find tractable solutions to the exploration-exploitation trade-off. In particular, I appreciated that the method was clearly positioned with respect to recent work on neural architecture search, meta-learning approaches to curiosity as well as forthcoming about the method’s limitations (outlining many hand-designed curiosity objectives which fall outside of their search space). There are also some interesting results in the appendix which show the efficacy of their predictive approach to program performance.

My main reservation is with respect to the empirical validation. Very few existing approaches to meta-learning curiosity scale to long temporal horizons and “extreme” transfer (where meta-training and validation environments are completely different). As such, there is very little in the way of baselines. The paper would greatly benefit from scaled down experiments, which would allow them to compare their architecture search approach to recent approaches [R1, R2], black-box optimization methods in the family of evolution strategies (ES, NES, CMA-ES), Thompson Sampling [R3] or even bandits tasks for which Bayes-optimal policies are tractable (Gittins indices). These may very well represent optimistic baselines but would help better interpret the pros and cons of using neural architecture search for meta-learning reward functions versus other existing methods. Conversely, the paper claims to “search over algorithms which [...] generalize more broadly and to consider the effect of exploration on up to 10^5, 10^6 timesteps” but at the same time does not attempt to show this was required in achieving the reported result. Pushing e.g. RL2 or Learning to RL baselines to their limits would help make this claim.

Along the same line, it is regrettable that the authors chose not to employ or adapt an off-the-shelf architecture search algorithm such as NAS [R4] or DARTS [R5]. I believe the main point of the paper is to validate the use of program search for meta-learning curiosity, and not the details of the proposed search procedure (which shares many components with recent architecture search / black-box optimization algorithms). Using a state-of-the-art architecture search algorithm would have made this point more readily.

Another important point I would like to see discussed in the rebuttal, is the potential for cherry-picking result. How were the “lunar lander” and “acrobot” environments (same question for “ant” and “hopper”) selected? From my understanding, it is cheap to evaluate learnt curiosity programs on downstream / validation tasks. A more comprehensive evaluation across environments from the OpenAI gym would help dispel this doubt. Another important note: top-16 results reported in Figure 4 and Table 1 are biased estimates of generalization performance (as they serve to pick the optimal pre-trained curiosity program). Could the authors provide some estimate of test performance, by e.g. evaluating the performance of the top-1 program (on say lunar lander) on a held-out test environment? Alternatively, could you comment on the degree of overlap between the top 16 programs for acrobot vs lunar lander? Thanks in advance.

[R1] Learning to reinforcement learn. JX Wang et al..
[R2] RL2: Fast Reinforcement Learning via Slow Reinforcement Learning. Yan Duan et al.
[R3] Efficient Bayes-Adaptive Reinforcement Learning using Sample-Based Search. Guez et al.
[R4] Neural Architecture Search with Reinforcement Learning. Barrett et al.
[R5] DARTS: Differentiable Architecture Search. Liu et al.


**Experience Assessment:**

I have read many papers in this area.

**Review Assessment: Checking Correctness Of Derivations And Theory:**

N/A

**Review Assessment: Checking Correctness Of Experiments:**

I assessed the sensibility of the experiments.

**Review Assessment: Thoroughness In Paper Reading:**

I read the paper at least twice and used my best judgement in assessing the paper.

---

> ### Author Response · Authors · 2019-11-15
> **Thank you for your detailed review!**
>
> We compared against human algorithms instead of previous meta-learning algorithms because human algorithms are a much stronger baseline. We go over this decision in more detail in a separate answer for all reviewers.
>
> The reviewer is right that we could have used many techniques from the existing literature to find good algorithms, such as RL as in [R4] or mentioned techniques from bandit settings to decide which program to evaluate next. We chose to keep the algorithm search as simple as possible and limit the number of operations for two reasons.
> 1. Since (to the best of our knowledge) we are the first to meta-learn learning algorithms (instead of network weights or network architectures) we wanted to make sure we gained an understanding of the problem setting. For instance, by evaluating a big part of the space of programs up to a certain size, we realized that program performance doesn’t form a continuum, but instead consists of about 1% of programs performing statistically much better than the statistically-indistinguishable set of all other programs.
> 2. Limiting the size of the programs in our search space allowed us to better interpret the best programs found by our search, which is useful to add confidence to our experimental results and gain algorithmic insight. Improving the efficiency of our search by including insights from other fields is an interesting avenue for future work.
>
> It is worth noting, however, that many techniques in the NAS community do not apply to search in algorithm space. There are three main challenges: first, curiosity algorithms form a dynamical system that intertwines with the RL agent and the RL environment. Therefore, NAS algorithms that reuse learned weights from previously tried architectures, such as [R5], are not immediately applicable because we want an algorithm that helps the RL agent learn from scratch. Second, many NAS algorithms (such as [R5]) assume each individual architecture is end-to-end differentiable, which is not the case for our algorithms. Finally, in NAS, the goal and loss function are the same for all architectures, which means that all substructures are likely to have similar representations and weights. In contrast, our algorithms also define their own optimization function, which strongly affects the composability of the substructures.
>
> We now describe how we chose the environments we ran on, for which we never considered the performance of our own algorithms. Gridworld was first selected because it measured exploration and was very simple and very cheap to run compared to other movement based environments like DeepMind labs or VizDoom. Then, we selected two standard OpenAI Gym environments by focusing on environments that were cheap to run. If one explores https://gym.openai.com/envs, they will observe that most other environments take either an order of magnitude more compute, are very similar to our chosen environments or are about keeping something stable (CartPole), for which curiosity is clearly detrimental. For all 3 environments, we chose the number of time-steps to train each RL agent by maximizing the difference between the performance of published works and our “dumb” baselines of fixed or pure-noise rewards.
>
> We then moved to MuJoCo tasks because they were an order of magnitude more expensive (but not two orders, like Atari games) and tested 3 environments: Ant, Hopper and Walker2d, the latter was discarded because published methods didn’t statistically outperform the dumb curiosity algorithm baselines. For Ant and Hopper we again selected the number of training time-steps to run by maximizing the distance between published results and the weak baselines. Therefore, we believe environment selection helped the strong baselines more than our algorithms. We were, unfortunately, unable to test on more environments because running thousands of agents on a new environment takes a significant portion of our compute budget.
>
> Finally, we note that LunarLander and Acrobot are also held-out test environments. Our initial intention was indeed to use LunarLander and/or Acrobot for meta-training; however, when we saw that the top programs in Gridworld did great in the other environments, we decided it was a much stronger message to show that we only needed a single very simple task as meta-training to find good algorithms. Therefore, what we refer to as top-16 is always the same set of programs found in Gridworld. We then show they also perform well in Lunar Lander, Acrobot and MuJoCo; even if the environments are very different.

---

### Author Response · Authors · 2019-11-15
**General comment to all reviewers**

We thank the reviewers for their useful comments.

*** Overview of changes in our submission ***
- You can find our cleaned code here: http://bit.ly/meta-learning-curiosity-algs
- We added the performance of baselines for GridWorld, Acrobot and LunarLander in figure 4; showing that top programs found on GridWorld have equivalent performance on LunarLander and significantly better performance on Acrobot. We also would like to note that we chose to report average rewards instead of final performance. Had we reported the latter, results for both the baselines and our algorithms would be much higher, but also a lot more noisy, which would have required a lot more episodes (and thus compute) to get statistical significance on thousands of programs.
- We added more details on how we predict program performance, including a plot in appendix C comparing predicted versus actual performance.

*** Discussion of our benchmarks and competing meta learning approaches ***
We would like to point out that we missed a relevant paper in our related work: Evolved policy gradients (EPG) by Houthooft et al. (https://arxiv.org/pdf/1802.04821.pdf). EPG meta-learns a loss function (similar to our reward function) that helps agents train from scratch to achieve a certain goal. While they meta-learn the parameters of a neural network that computes the loss function, we instead meta-learn interpretable learning algorithms. This allows us to generalize in much broader ways.

EPG shares our goal of increasing meta-learning generalization. However, their generalization is limited to different targets within a single environment. They meta-train their algorithm to move to eastward target locations, and then meta-test on westward target locations; showing that MAML and RL^2 fail to adapt to the new task but EPG succeeds in doing so. We believe this type of generalization stands in stark contrast to our programs that generalize to completely new environments, with different action dimensions, continuous action spaces and even from image inputs to vector inputs.

Transferring parameters (such as in MAML, RL^2 or EPG) between environments with different observation and action spaces is challenging. Dimensionality transfer can be achieved through summary statistics, random embeddings, etc. but these hacks heavily constrain the type of functions that can be learned. Transferring between vector-based observation spaces and image-based observation spaces is even more problematic.  In contrast, our algorithms instantiate networks of the proper type and dimensionality automatically depending on the environment.

Rather than comparing against approaches not designed to transfer parameters across environments, we decided to compare against human-designed algorithms. These algorithms were designed for general environments and we believe this makes them significantly stronger competitors. Since our initial submission, we have verified this belief by designing and training a baseline variant of EPG to meta-learn across environments. Due to varying environment I/O specs, we encoded states into fixed-length vectors through randomly initialized neural networks. This variant of EPG performed significantly worse than the human-designed curiosity algorithms that we used as challenging baselines.

---

### Decision · Program_Chairs · 2019-12-19

**Decision:**

Accept (Poster)

**Comment:**

This paper proposes meta-learning auxiliary rewards as specified by a DSL. The approach was considered innovative and the results interesting by all reviewers. The paper is clearly of an acceptable standard, with the main concerns raised by reviewers having been addressed (admittedly at the 11th hour) by the authors during the discussion period. Accept.